# Two Ways to Achieve the Same Goal—Two Validated Quantitative NMR Strategies for a Low-Abundance Natural Product in Standardized Extracts: The Case of Hepatodamianol in *Turnera diffusa*

**DOI:** 10.3390/molecules27196593

**Published:** 2022-10-05

**Authors:** Aída Parra-Naranjo, Cecilia Delgado-Montemayor, Ricardo Salazar-Aranda, Rocío Castro-Ríos, Alma L. Saucedo, Noemí Waksman-Minsky

**Affiliations:** 1Facultad de Medicina, Departamento de Química Analítica, Universidad Autónoma de Nuevo León, Monterrey 64460, Mexico; 2Consejo Nacional de Ciencia y Tecnología, Ciudad de México 03940, Mexico

**Keywords:** qNMR, method validation, nuclear magnetic resonance, natural products, damiana

## Abstract

The quantification of low-abundance secondary metabolites in plant extracts is an analytical problem that can be addressed by different analytical platforms, the most common being those based on chromatographic methods coupled to a high-sensitivity detection system. However, in recent years nuclear magnetic resonance (NMR) has become an analytical tool of primary choice for this type of problem because of its reliability, inherent simplicity in sample preparation, reduced analysis time, and low solvent consumption. The versatility of strategies based on quantitative NMR (qNMR), such as internal and external standards and electronic references, among others, and the need to develop validated analytical methods make it essential to compare procedures that must rigorously satisfy the analytical well-established acceptance criteria for method validation. In this work, two qNMR methods were developed for the quantification of hepatodamianol, a bioactive component of *T. diffusa*. The first method was based on a conventional external standard calibration, and the second one was based on the pulse length-based concentration determination (PULCON) method using the ERETIC2 module as a quantitation tool available in TopSpin software. The results show that both procedures allow the content of the analyte of interest in a complex matrix to be determined in a satisfactory way, under strict analytical criteria. In addition, ERETIC2 offers additional advantages such as a reduction in experimental time, reagent consumption, and waste generated.

## 1. Introduction

*Turnera diffusa* (commonly called damiana) is a shrub that grows in the arid and semiarid regions of South America, Mexico, and USA; its leaves are small and wrinkled with a strong odor and yellow flowers. In traditional medicine, it has been used for the relief of colds and coughs, among other conditions [1]. *T. diffusa* is considered the most important species of the turneraceae; evidence of this is the growing number of scientific publications focused on the study of its therapeutic effects as well as in the quality control of herbal products [2,3,4,5,6,7,8,9]. Numerous research groups have reported its biological activities, including its use as an aphrodisiac [10], antidiabetic [11], antioxidant [5], amidst others. Among its biological activities, the hepatoprotective activity stands out, which our work group demonstrated both in in vitro as well as in vivo models [12]. The preliminary results show that the methanolic extract from *T. diffusa* is associated with an antifibrotic effect by decreasing profibrotic and mitochondrial markers. Furthermore, it was suggested a probable mechanism by which the extract could perform its hepatoprotective role is by inducing apoptosis of activated hepatic stellate cells [13]. Through biodirected isolation, we were able to identify hepatodamianol (luteolin 8-C-[6 deoxy-2-O-(alpha-L-rhamnopyranosyl)-xyl-hexopyranos-3-uloside]) as the main compound responsible for the hepatoprotective activity [14] (Figure 1, Structure **1**). Hepatodamianol is deemed a biomarker [7] because, to date, it is only found naturally in this plant, which is why we consider the quantitation of this metabolite important for the quality control of products based on *T. diffusa*. The lack of commercially reliable hepatodamianol standards, the low concentration of hepatodamianol in the plant as well as the difficulty of its isolation make it difficult to have a standard of this compound in adequate amounts for the development and validation of an analytical method. Therefore, it is necessary to explore new strategies to perform the quantitative analysis of this metabolite in extracts and herbal drugs.

In this sense, nuclear magnetic resonance (NMR) represents an alternative. Historically, NMR has been related to natural product analysis, mainly regarding structure elucidation. However, NMR is a universal and intrinsically quantitative analytical spectroscopic technique, and recently, it has emerged as a powerful tool for metrological applications with unique capabilities [15]. For these reasons, proton quantitative NMR (^1^H-qNMR) has been successfully applied in the analysis of natural products in complex matrices such as botanicals, supplements, and traditional herbal medicines [16]. In fact, despite its inherent sensitivity limitation, ^1^H-qNMR also offers multiple calibration options [17,18], some of which do not require the use of standards identical in nature to the analyte [19]. Among these options, the pulse length–based concentration determination (PULCON) method stands out because it is a methodology based on the use of a second resonance signal generated by an external standard serving as reference. In the PULCON method, the area obtained by the integration over a resonance in the spectrum of a reference sample is correlated with the area of a resonance with unknown concentration. Then, when the external standard signal is calibrated, it allows the quantitation through the reciprocity principle which states that the intensity of an NMR signal is inversely proportional to the duration of the 90° pulse [20]. The PULCON methodology has been successfully employed in the evaluation of the purity of pharmaceutical reference materials [21], for the quantitation of metabolites in serum [22] and cell cultures [23], taurine in energy drinks [24], and some natural products [25,26,27,28]. The module ERETIC2, available in TopSpin software (Bruker, Germany), is a quantitation tool based on PULCON that can be used for quantifications from ^1^H-NMR spectra. Based on this, the objective of this work was to develop and validate a quantitation method for hepatodamianol in standardized extracts of *T. diffusa*, using two calibration modalities to compare its performance, PULCON through ERETIC2 and the external standard calibration method using rutin (Figure 1, Structure **2**) as calibrant. 

## 2. Results and Discussion

### 2.1. Selection of Solvent and Signal for Quantitation

NMR can be used in the quantitative analysis of complex mixtures as long as a characteristic signal of the analyte is available and well-resolved in the ^1^H spectrum. In the ^1^H-NMR spectrum of hepatodamianol, Figure 2A, the hydroxyl proton at δ_H_ 13.18 ppm (1H, m, OH-5), the xylo-hexopyranos-3-uloside methyl group at δ_H_ 1.40 ppm (3H, d, J = 5.8 Hz, H-6″), and the rhamnose methyl group at δ_H_ 0.515 ppm (3H, d, J = 6.2 Hz, H-6‴) are readily distinguishable signals. However, the spectrum obtained from the standardized extract, Figure 2B, shows a complex appearance with few candidate signals for use in quantitation. In fact, the signal at δ_H_ 0.51 ppm stands out, which is well-resolved in both the standardized extract and hepatodamianol spectra. Thus, this signal, which corresponds to CH_3_-6‴, was chosen to be used for quantitation. Due to the high solubility of the extract in DMSO-*d*_6_, it was chosen as the solvent for the analysis.

### 2.2. Purity Evaluation of the Isolated Hepatodamianol

The evaluation of the purity of hepatodamianol isolated from *T. diffusa* was performed using different experimental procedures. First, the chromatographic purity was calculated using the internal normalization method, which consists of calculating the % area of the peak of interest in relation to the total area of the peaks present in the chromatogram. Second, the spectroscopic purity was obtained by NMR through two calibration methods. For this purpose, the PULCON method available in the TopSpin software through the module ERETIC2 was used. This methodology is based on the work undertaken by Wider and Drier [20]. It consists of generating a second resonance signal, which is calibrated with a reference solution. Under this experimental scheme, it is possible to calculate the molar concentration of the analyte in the solution through the following equation: (1)$$C_{unk}=kC_{ref}\times \left(\frac{A_{unk}}{A_{ref}}\right)\times \left(\frac{T_{unk}}{T_{ref}}\right)\times \left(\frac{\theta_{unk}}{\theta_{ref}}\right)\times \left(\frac{n_{ref}}{n_{unk}}\right)$$
where the subscripts unk and ref refer to the analyte (of unknown concentration) and reference substances, respectively; *C* indicates the molar concentration; *A* is the signal integration value; *T* is the temperature; *θ* is the pulse length; n is the number of protons generating the resonance; and *k* refers to a correction factor of the receiver gain (rg). However, when rg is kept constant then *k* is equal to 1. As the temperature was controlled and fixed to 298 K during the acquisition of the NMR data of reference as well for the analyte, this term is equal to 1. The same occurs with the pulse angle term because it was fixed to 90° in both cases. After these reductions, Equation (1) can be simplified and written as:(2)$$C_{unk}=C_{ref}\times \left(\frac{A_{unk}}{A_{ref}}\right)\times \left(\frac{n_{ref}}{n_{unk}}\right)$$

In the present work, a 5 mM benzoic acid certified standard was used to calibrate the ERETIC2 signal. The spectroscopic purity was also calculated through a calibration curve with rutin as an external standard. As qNMR is considered a primary analytical method, the reference materials do not need to be identical in nature or even be chemically related to the analyte. Due to the lack of reliable commercially available hepatodamianol standards and the technical difficulty and low yield in which hepatodamianol is obtained through isolation, we chose to use rutin as an external standard. Structurally, rutin and hepatodamianol are similar; both are glycosylated flavonoids with a rhamnose moiety in their structure. The ^1^H NMR spectrum of rutin in DMSO-*d*_6_ shows a doublet at a δ_H_ 0.990 ppm (Figure 3); this signal corresponds to the CH_3_-6‴ of the rhamnose moiety. As the quantitation signal for hepatodamianol also corresponds to CH_3_-6‴ of the rhamnose portion, it was decided to use this same signal for the calibration curve with rutin as an external standard. Table 1 shows the results obtained for the calculation of the purity of the isolated hepatodamianol. When analyzing the results, an important difference between the chromatographic and the spectroscopic purities is observed. The results obtained are in agreement with those reported in the literature for comparisons between purity values measured by qNMR and chromatographic techniques, where there are discrepancies of between 10 and 20% in the results between these two techniques, with spectroscopic purity always being lower [29,30,31]. Although the result of the chromatographic purity of hepatodamianol and that of spectroscopic purity are significantly different, this is not the case with the spectroscopic purity calculated with the calibration curve and that obtained by PULCON. The agreement between these two results reinforces the usefulness of PULCON through ERETIC2 as an alternative quantitation method.

### 2.3. Measurement of Longitudinal Relaxation Time

To establish the best acquisition parameters for the ^1^H-qNMR spectra and to assure quantitative conditions, it was necessary to measure the longitudinal relaxation time (*T*_1_) of the hydrogens involved in the quantitation procedure. *T*_1_ is important for the optimization of the appropriate delay time between pulses, as it is recommended to set this value as five times the longest *T*_1_. This ensures that the magnetization has been fully reestablished [17] before the next 90° pulse application. Table 2 shows the values obtained for the analyzed nuclei. The *T*_1_ values of hydrogen are normally in the range of 0.3 to 5 s. Based on the obtained results, the d1 time was set to 3 s, which corresponds to 5 times the longest *T*_1_, corresponding to CH_3_-6‴ of hepatodamianol.

### 2.4. Method Validation

The method performance was evaluated according to criteria established in the Guide to NMR Method Development and Validation—Part 1: Identification and Quantification [32]. In this procedure, general conditions for NMR analysis are described (such as acquisition and processing parameters) as well identification criteria and qNMR parameters for validation. In the following sections, validation parameters for hepatodamianol quantitation are reviewed.

#### 2.4.1. Specificity

Specificity means the ability to assess unequivocally the analyte of interest in the presence of other components; then, the specificity in an NMR method can be assessed through the unambiguous assignment of all NMR resonances to the structure of the analyte [33]. Furthermore, the quantitation signal should not overlap with other signals. Sometimes it is adequate to observe the ^1^H spectrum to identify overlaid signals, but other types of experiments can also be used, such as 2D or correlation experiments [34]. In the present work, specificity was evaluated through the selective 1D-TOCSY experiment, which unequivocally ensures that the quantitation signal corresponds to the analyte; this is because it is not only one signal that is taken as confirmation of identity but rather the entire signal pattern of the spin system involved. To confirm the usefulness of the 1D-TOCSY experiment as a selectivity parameter, an experiment was first performed with rutin. The selective 1D-TOCSY spectrum obtained from rutin shows the signal pattern corresponding to the rhamnose moiety coupled to CH_3_-6‴ protons, which allowed us to assess the presence of this compound in a sample and demonstrate the selectivity of CH_3_-6‴ resonance for quantitative purposes (data not shown). In the same way, hepatodamianol contains a 2-O-alpha-L-rhamnopyranoside moiety, where the hydrogens 1‴, 2‴, 3‴, 4‴, 5‴, and 6‴ form a spin system. Although the doublet signal of CH_3_-6‴ is easily identified in the ^1^H spectrum of the standardized extract (Figure 2B), this is not the case for the other protons that are part of the CH_3_-6‴ spin system. 

Similar to rutin, when the selective 1D-TOCSY spectrum is recorded for hepatodamianol:CH_3_-6‴ resonance, the spin system can be clearly distinguished. Figure 4A shows the selective 1D-TOCSY spectrum of hepatodamianol, and Figure 4B shows that of a standardized extract sample. As is evident, the complexity of the spectrum is significantly reduced because only the signals of the hydrogens coupled in the spin system of the irradiated nucleus are observed. In the selective 1D-TOCSY spectrum, the six signals of the spin system conforming the hepatodamianol rhamnopyranoside moiety are clearly identified (CH-1‴, δ_H_ 4.60 ppm; CH-2‴, δ_H_ 3.65 ppm; CH-3‴, δ_H_ 3.00 ppm; CH-4‴, δ_H_ 2.94 ppm; CH-5‴, δ_H_ 2.12 ppm; and CH_3_-6‴, δ_H_ 0.51 ppm), and the chemical shift values agree with those previously reported for this molecule [8]. This evidence indicates that the signal at 0.515 ppm corresponds unequivocally to hepatodamianol:CH_3_-6‴.

#### 2.4.2. Linearity

To evaluate the linearity of the method, a calibration curve was constructed, and a regression analysis was performed. For the external standard calibration (ESC) method with rutin, the area of the CH_3_-6‴ signal from this compound was plotted against the concentration of the calibration standards, whereas for the PULCON calibration method, the area of the same signal was plotted against the concentration calculated using the PULCON program. Table 3 summarizes the results obtained for the linearity. The results obtained by both calibration methods show a linear behavior in the concentration range evaluated, with r^2^ values > 0.999 for both calibration modalities.

#### 2.4.3. Precision

Precision was assessed with the percent relative standard deviation (%RSD) of the calibration standard response (area). This calculation was carried out for each set of triplicates of the rutin standard solutions. The interday precision was evaluated with 0.50, 2.50, and 5.00 mM solutions. The intraday precision results are tabulated in Table 4, and the interday precision results are tabulated in Table 5. The obtained %RSD values for both calibration modes are similar. Compared with chromatographic methods developed for the quantitation of hepatodamianol where the %RSD is up to 2% for intraday precision and up to 8% for interday precision [7], the precision of the qNMR method proposed in this work, either through calibration by external standard or by PULCON, offers superior results.

#### 2.4.4. Accuracy

Accuracy was evaluated with the percentage error (% Error), using the following formula:(3)$$\%\;Error:\;\frac{calculated\;concentration-theorical\;concentration\;}{theorical\;concentration}\times 100$$

The theoretical concentration of the standard solutions and the concentration obtained with the calibration curve equation were used for the calculation. Table 4 shows the results obtained, both for the calibration by external standard with rutin and for the ERETIC2 method. For both calibration methods, the highest % Error is found at the lowest concentration level, which is to be expected due to the lower signal-to-noise (S/N) ratio obtained at low concentrations. Furthermore, the recovery was evaluated by adding a known concentration of hepatodamianol (250 μL, 2.5 mM hepatodamianol) to a standardized extract sample, and the percentage recovery was then calculated. Figure 5 shows the overlaid spectra of the unspiked and spiked samples and indicates that the CH_3_-6‴ signal is increased. Table 6 shows the recovery results obtained by both calibration modalities; comparable results are obtained, with recovery in the ERETIC2 method being slightly lower but without significant difference (*p*-value > 0.05). Nevertheless, the percentage recovery values obtained are in agreement with those reported for qNMR methods applied to natural products [35,36,37].

#### 2.4.5. Limit of Detection

The limit of detection (LOD) is defined as the minimum concentration at which the analyte can be reliably detected. In NMR methods, it is desirable that this concentration allows us to distinguish the signal pattern corresponding to the analyte [32]. As both the quantitation signal of hepatodamianol and the rutin signal used for the construction of the calibration curve correspond to the CH_3_-6‴ group of the rhamnose moiety in both structures, it is expected that both signals present the same shape, intensity, and integration value at a given concentration level. Based on this, the LOD was set as the minimum concentration at which the rutin signal pattern corresponding to CH_3_-6‴ (δ_H_ 0.99 ppm), CH-1‴ (δ_H_ 4.39 ppm), OH-4″ (δ_H_ 5.08 ppm), CH-1″ (δ_H_ 5.35 ppm), CH-5′ (δ_H_ 6.85 ppm), CH-2′ (δ_H_ 7.53 ppm), and OH-5 (δ_H_ 12.60) is identified. Figure 6 shows rutin spectra at concentrations of 0.25, 0.125, and 0.10 mM, and the above signals are indicated. The LOD was set at 0.125 mM because the referred signals are not fully distinguishable below this concentration.

#### 2.4.6. Limit of Quantitation

The limit of quantitation (LOQ) is defined as the minimum concentration of analyte that can be quantitatively determined with adequate precision and accuracy. Table 7 shows the %RSD and % Error obtained for both the calibration curve and ERETIC2 analysis of rutin at 0.125 and 0.250 mM. Although the 0.125 mM concentration generated results with an acceptable %RSD, the % Error was significantly greater for both the calibration curve and ERETIC2 analyses. Based on these results, the LOQ was set at 0.250 mM, a concentration at which %RSD and % Error were <5 (absolute).

#### 2.4.7. Robustness

In this analytical procedure, robustness was evaluated by modifying (a) the line broadening (LB) value (0.05–0.15 Hz), (b) spectra processing by three different analysts (A1–A3), and (c) the baseline correction algorithm employed (ABS, ABSN, and ABSD). Table 8 shows the average analyte concentration obtained and the %RSD of each parameter considered to evaluate the robustness at three concentration levels. As is evident, there are no significant differences in the performance of the method when analyzed with external standard calibration and PULCON. It has been reported that one of the factors that most affects the quality of the results obtained by qNMR is the processing of the data by different analysts [38]; that is, aspects such as integration and phase correction can be subjective if the analysis is performed by different analysts. In this work, we demonstrate that the method developed is suitable for the processing of spectra by different analysts at the three concentration levels considered.

Likewise, the method was also robust for the LB parameter at all three concentration levels evaluated. Monakhova et al. [39] report that qNMR results are not influenced by changing the LB value in the range 0.10–0.80 Hz. In the present work, the changes were made at 0.05, 0.10, and 0.15 Hz. Regarding the baseline correction algorithm parameter, no significant difference was found between the results obtained with the three algorithms evaluated, both for the medium and high levels of concentration; however, differences were observed for the low concentration level. 

The general way in which baseline correction algorithms work involves three steps. First, it identifies which information in the spectrum corresponds to signals and which information corresponds to noise. This information is then used to build a baseline model to finally correct the signals in the spectrum by subtracting the baseline model from the original signals [40]. Normally, to identify which sections of the spectrum correspond to resonance signals, the baseline correction algorithms calculate the standard deviation of the noise. This offers an explanation for the finding that the quantitation method is not robust for low concentration levels where the S/N ratio is lower, and it generates significant differences between what is considered noise and what is considered the external standard calibration signal.

### 2.5. Hepatodamianol Quantitation in Standardized Extract Samples

The analytical method here described was designed as an alternative procedure for the quantification of hepatodamianol in standardized *T. diffusa* extracts. Table 9 presents the results obtained for the quantitation of hepatodamianol in four different samples of standardized extracts of *T. diffusa*, obtained by a method recently reported by us [12]. Although the %RSD values are slightly higher for the ERETIC2 method compared with those obtained using the calibration curve, the difference in the average concentration calculated by both methods is not statistically significant (*p* > 0.05).

## 3. Experimental: Materials and Methods

### 3.1. Chemicals and Reagents

Hexadeuterated dimethyl sulfoxide (DMSO-*d*_6_, 99.8% D, TMS 0.05%) and deuterated methanol (CD_3_OD-*d*_4_, 99% D) were purchased from Sigma-Aldrich (St. Louis, MO, USA). Certified standards of benzoic acid (5.01 ± 0.01 mM) and rutin trihydrate (97.67%) were purchased from Cambridge Isotope Laboratories (Cambridge, UK) and HWI Analytik GmbH (Rülzheim, Germany), respectively. Hepatodamianol was isolated from *T. diffusa* following a previously reported method [7]. All solvents used for extraction and purification (methanol, ethylacetate, acetic acid) were of analytical grade and purchased from Fermont (Monterrey, Nuevo León, Mexico). Deionized water was obtained from an Elga II water purification system (Veolia, UK).

### 3.2. Plant Material

Four *T. diffusa* specimens (samples 1–4) were collected in Montemorelos, Nuevo León (25.187° N, 99.826° W) between 2014 and 2016. These plants were authenticated by a voucher specimen of *T. diffusa* (No. 23569) that was deposited in the herbarium at the Facultad de Ciencias Biológicas, Univesidad Autónoma de Nuevo León. Aerial parts of plants were dried at room temperature until constant weight, prior to grinding. Samples were stored in darkness until required for the extraction procedure. 

### 3.3. Sample Preparation

The methodology followed to obtain the standardized extracts was previously described by Delgado-Montemayor et al. [12]. Briefly, 30 g of air-dried *T. diffusa* ground plant was extracted using methanol maceration at room temperature (3 × 500 mL, 2 h, 300 rpm) in a Heidolph Unimax 1010 orbital shaker (Thermo Fisher Scientific, Waltham, MA, USA). All the extracts were pooled together and evaporated to dryness under vacuum to yield a viscous mass. Chlorophyll content was eliminated using SEP-PAK C-18 cartridges (1000 mg/8 mL; Alltech Associates Inc., Deerfield, IL, USA). Samples were eluted with 50%, 70%, and 100% methanol fractions. Then, 5 g of the fraction obtained using 50% methanol was subjected to vacuum liquid chromatography (LC) on silica gel, using dichloromethane, ethyl acetate, ethyl acetate/methanol (1:1), and methanol as eluents (silica gel 60 G for thin-layer chromatography, Merck Millipore; 400 mL of each solvent). Finally, the obtained fractions were freeze-dried at 223 K and 0.133 mbar. For quantitative analysis, 25 mg of the ethyl acetate/methanol fraction were accurately weighed using an analytical balance (Ohaus Pioneer PA214C, USA) and dissolved with DMSO-*d*_6_ into a 1.0 mL class A volumetric flask, transferred to a conical plastic tube, and centrifuged for 5 min at 13,000× *g* at room temperature. Finally, 600 μL of deuterated solution was transferred to 5 mm standard NMR tubes for quantitative analysis. 

### 3.4. Evaluation of Purity of the Isolated Hepatodamianol Using High-Performance Liquid Chromatography with Diode Array Detection

High-performance liquid chromatography with diode array detection was performed as previously described [12] with minor modifications. A Waters Alliance 1525 LC system equipped with an online degasser, binary pump, autosampler, and a 2996 diode array detector (Waters, Milford, MA, USA) was used. The separation was carried on a Hypersil Gold C18 reversed-phase column (4.6 × 150 mm, 5 μm; Thermo Fisher Scientific, Waltham, MA, USA) in gradient mode with methanol and aqueous 0.1% formic acid solution as mobile phases at a flow rate of 0.4 mL/min. The elution program commenced with 30% methanol, changing to 60% methanol in 20 min with a linear gradient, then increasing to 70% methanol in 5 min, held for 5 min, and subsequently returned to the initial conditions in 10 min with an equilibration time of 5 min. The injection volume was 10 μL. Samples were always analyzed in duplicate. Data acquisition in the diode array detector was recorded from 210 to 400 nm with a resolution of 1.2 nm per diode. Hepatodamianol quantitation was carried out at 254 nm.

### 3.5. Evaluation of the Purity of Isolated Hepatodamianol Using qNMR

The isolated hepatodamianol was weighed in triplicate using an analytical balance (±0.1 mg; Ohaus Pioneer) and dissolved in DMSO-*d*_6_ using 1.0 mL class A volumetric flasks. The identity of the isolated hepatodamianol was confirmed using ^1^H and ^13^C routine experiments in 1D and 2D (^1^H, ^13^C, DEPT 135, COSY, HSQC, and HMBC). The hepatodamianol concentration was calculated using the ERETIC2 tool in TopSpin (v 3.6.1), based on the PULCON method, using a certified standard solution of benzoic acid (5.01 ± 0.01 mM) as reference; proton resonances at 7.945 (dd), 7.624 (dt), and 7.501 (dd) ppm were manually integrated to obtain the average area per proton. The hepatodamianol signal used for quantitation was the doublet corresponding to the CH_3_-6‴ protons (δ_H_ 0.51 ppm). The unknown concentration (*C_unk_*) of the analyte was calculated as follows:(4)$$C_{unk}=C_{ref}\frac{A_{unk}n_{ref}}{A_{ref}n_{unk}}$$
where *C_ref_* is the benzoic acid concentration; *A_unk_* is the area of the CH_3_-6‴ peak; *n_ref_* refers to number of protons generating the resonance, in this case 1; *A_ref_* is the benzoic acid average area per proton; and *n_unk_* is equal to 3 (the number of protons in the CH_3_-6‴ methyl group).

### 3.6. Preparation of Standard Solutions 

A stock solution of rutin trihydrate was prepared to a final concentration of 10.29 mM in DMSO-*d*_6_. To build a calibration curve, a set of rutin solutions ranging from 0.50 to 5.00 mM were prepared in triplicate using 1.0 mL class A volumetric flasks. Another series of dilutions ranging from 0.25 to 0.10 mM were prepared to determine the LOQ and the LOD. To evaluate the accuracy of the method, a solution of hepatodamianol in DMSO-*d*_6_ was prepared with a final concentration of 2.50 mM. This solution was also used to perform experiments to determine percentage recovery. The rutin resonance used as calibrant was the doublet corresponding to CH_3_-6‴ (δ_H_ 0.992 ppm).

### 3.7. NMR Experiments

NMR experiments were recorded on a 400 MHz Avance III spectrometer (Bruker) equipped with a 5 mm BBO SmartProbe with a z-gradient pulse field. Proton NMR spectra were recorded using a 90° pulse experiment, under the following acquisition parameters: 16 scans with a fixed rg value of 32; spectral width, 20.0 ppm; 65,536 points in time domain; and acquisition time, 4 s. Relaxation delay was 3 s, which was adjusted after performing the inversion recovery pulse sequence protont1 to determinate the relaxation time *T*_1_. Pulse sequence selmlpg was used for acquisition of selective 1D-TOCSY spectra using the acquisition parameters previously reported by Lucio-Gutiérrez et al. [8]. All NMR data were acquired at 298 K, and chemical shifts were referenced to the tetramethylsilane (TMS) signal at 0.0 ppm. NMR data were processed and analyzed with TopSpin software (Bruker). Prior to Fourier transformation, an exponential weighting factor corresponding to a LB of 0.10 Hz was applied. Spectra were manually phased and baseline correction was performed with the ABSN algorithm. Integration was performed in a semiautomatic procedure, in the chemical shift range 1.022–0.960 ppm for rutin CH_3_-6‴ resonance and 0.560–0.497 ppm for hepatodamianol CH_3_-6‴ resonance.

### 3.8. Method Validation

The method was inspected for specificity, linearity, precision (evaluated as %RSD), accuracy (evaluated as the relative % Error and percentage recovery), LOQ, LOD, and robustness according to the Eurolab guidelines for NMR quantitation [32]. Specificity: specificity was evaluated by means of the selective 1D-TOCSY experiment. Linearity: rutin solutions were prepared in triplicate at 0.25, 0.50, 1.00, 2.00, 3.00, 4.00, and 5.00 mM. The relationship between the integration of the CH_3_-6‴ rutin signal and the corresponding concentration was determined using linear regression analysis. Precision: intraday precision was evaluated by analyzing three replicates of the rutin standard solutions employed for the calibration curve. Interday precision was assessed by analyzing three replicates of rutin solutions at 0.50, 2.00, and 5.00 mM on three nonconsecutive days. Accuracy: rutin standard solutions employed for the calibration curve were analyzed in triplicate to calculate the % Error. For the recovery tests, spiked (250 µL, 2.50 mM hepatodamianol) and unspiked solutions of *T. diffusa* extracts were analyzed and percentage recovery was calculated. LOQ: limit of quantitation was established as the lowest concentration with an accuracy error and precision of <5%. LOD: limit of detection was defined as the lowest concentration displaying the signal pattern of rutin. Robustness: robustness was evaluated by modifying experimental parameters such as the LB value (0.05 and 0.15 Hz), the baseline correction algorithm employed (ABS and ABSD), and spectra processing by different analysts.

### 3.9. Comparison of Calibration Approaches

To compare the results obtained by the ESC method with the PULCON calibration method, the samples analyzed in the validation were reprocessed with the ERETIC2 tool. A 5 mM benzoic acid standard was used to calibrate the signal in ERETIC2, considering the experimental conditions indicated above and using Equation 1 to determine the hepatodamianol concentration. The results obtained by both calibration methods were compared in terms of linearity, precision, accuracy, LOQ, and robustness. 

### 3.10. Statistical Analysis

Statistical analysis was carried out using Excel (Windows 10, Microsoft, Redmond, WA, USA). All analyses were performed in triplicate unless stated otherwise. The results presented are the averages of the obtained values, including the %RSD. Analysis of variance (ANOVA) tests were carried out to evaluate robustness. Student’s t test was performed to evaluate the difference between the results obtained with the calibration curve and with ERETIC2 for the recovery test and the hepatodamianol content. Data manipulation was performed using Excel (Microsoft). 

## 4. Conclusions

The analyses carried out in this work indicate that the performance of ERETIC2 based on PULCON methodology is comparable with other more traditional calibration methods, such as the ESC method. Thus, ERETIC2 has several advantages over calibration using an external standard; for example, the preparation of solutions for the calibration curve is unnecessary, which means savings in terms of time and resources. As long as the spectrum used to calibrate the ERETIC2 signal is acquired under the same conditions as the spectra where the quantitation will be performed, ERETIC2 represents an effective alternative as a calibration method.

## Figures and Tables

**Figure 1 molecules-27-06593-f001:**
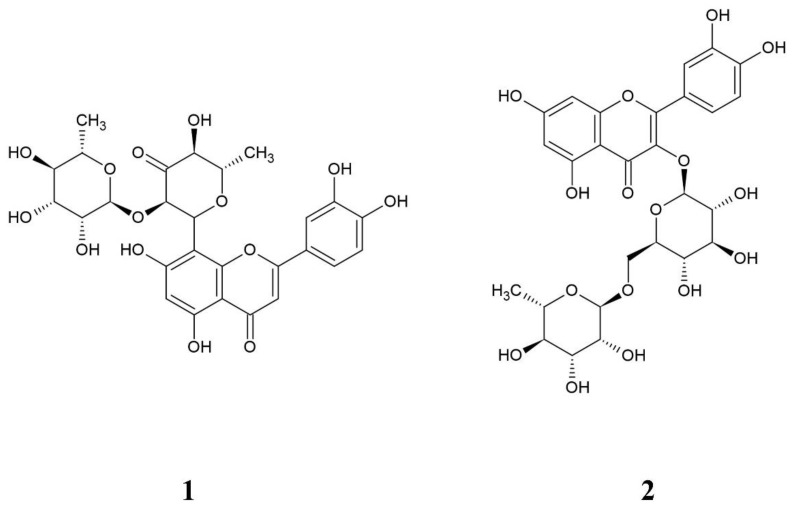
Chemical structure of hepatodamianol (**1**) and rutin (**2**).

**Figure 2 molecules-27-06593-f002:**
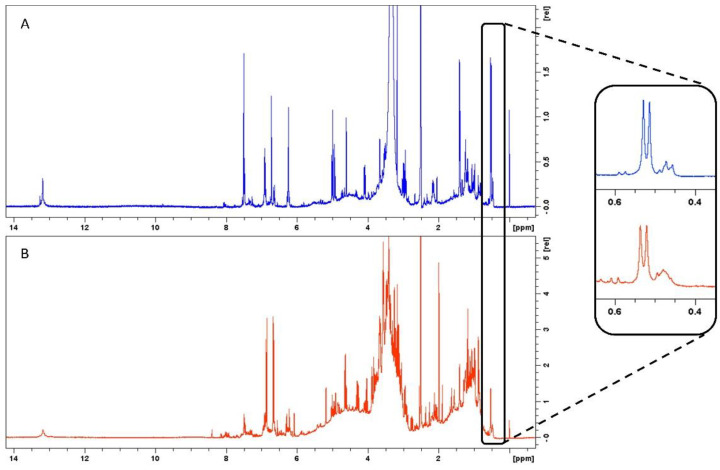
**(A**) ^1^H-NMR spectrum of hepatodamianol (400 MHz, DMSO-*d*_6_). (**B**) ^1^H-NMR spectrum of a standardized extract of *T. diffusa* (25 mg/mL, 400 MHz, DMSO-*d*_6_). The outside expansion shows the doublet resonance of hepatodamianol:CH_3_-6‴ in 0.515 ppm; it was used to quantitate this compound.

**Figure 3 molecules-27-06593-f003:**
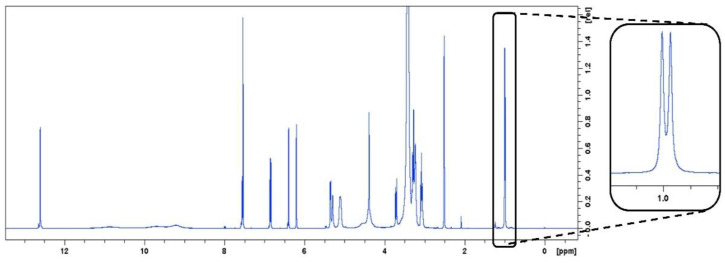
^1^H-NMR spectrum of rutin (400 MHz, DMSO-*d*_6_). The outside expansion shows the doublet resonance of rutin:CH_3_-6‴ in 0.990 ppm; it was used to quantitate this compound.

**Figure 4 molecules-27-06593-f004:**
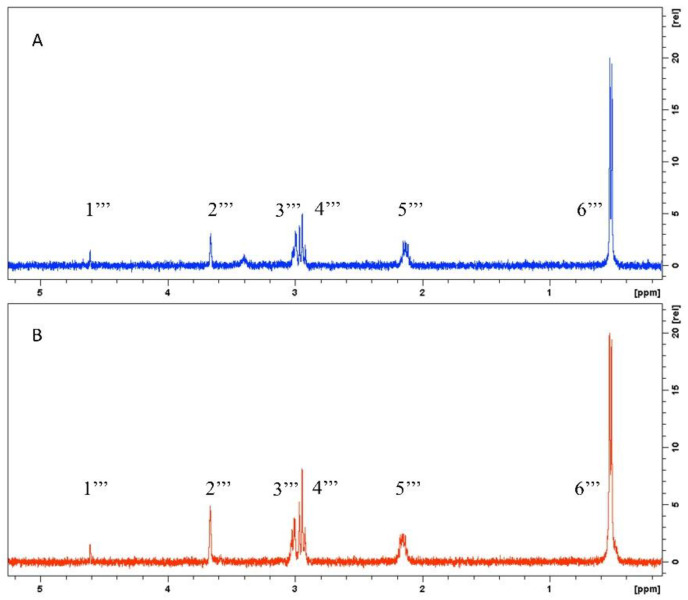
Selective 1D-TOCSY (400 MHz, DMSO-*d*_6_) of hepatodamianol:CH_3_-6‴. (**A**) 5 mM hepatodamianol standard solution. (**B**) *T. diffusa* standardized extract. The selective 1D-TOCSY spectrum allows us to assess the hepatodamianol content in *T. diffusa* standardized extracts and shows the selectivity of hepatodamianol:CH_3_-6‴ resonance in qNMR experiments.

**Figure 5 molecules-27-06593-f005:**
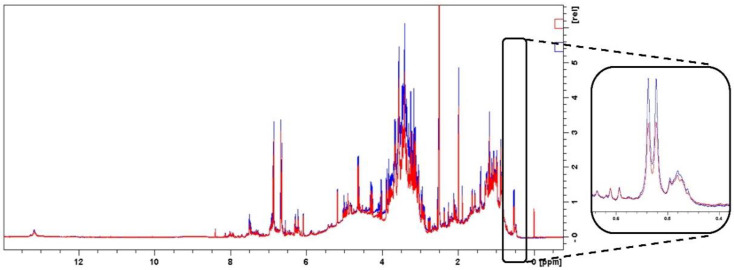
Overlapped ^1^H-NMR spectra (400 MHz, DMSO-*d*_6_) of *T. diffusa* standardized extract (red line) and *T. diffusa* standardized extract spiked with 250 μL of 2.5 mM hepatodamianol (blue line). The expansion shows the region of the observed hepatodamianol:CH_3_-6‴ resonance at 0.515 ppm.

**Figure 6 molecules-27-06593-f006:**
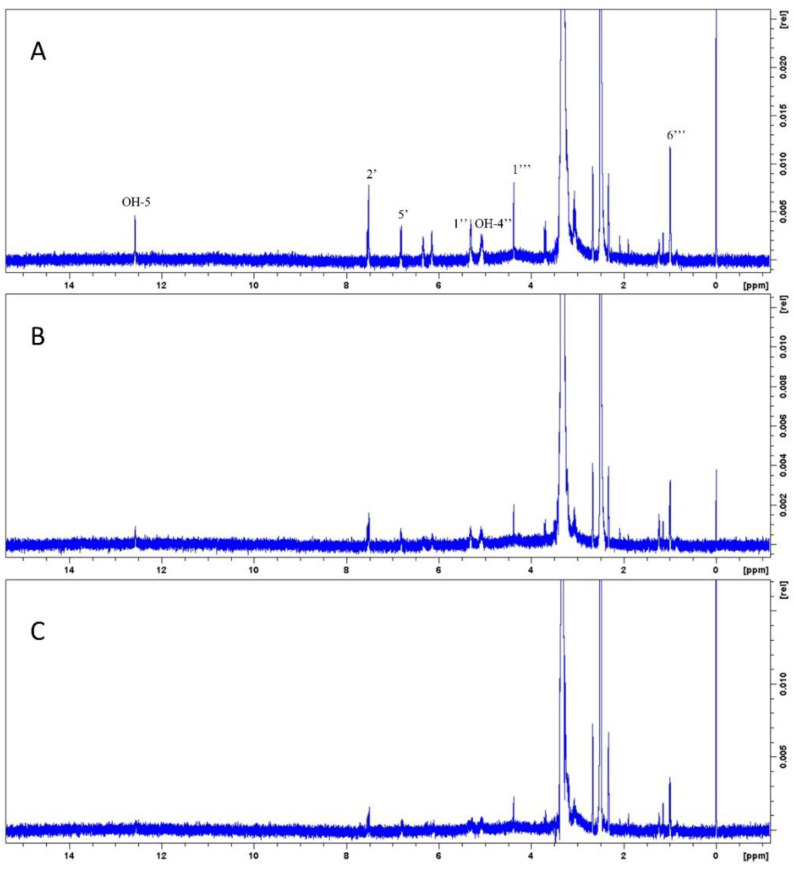
Experimental determination of the limit of detection (LOD) through a ^1^H-NMR spectrum (400 MHz, DMSO-d_6_). Rutin concentration: (**A**) 0.250 mM, (**B**) 0.125 mM, and (**C**) 0.100 mM. As concentration decreases, the distinctive molecular profile vanishes; then, the LOD is the lowest concentration at which the relevant signals in the spectrum can be visually distinguished. The rutin signals are clearly seen at 0.125 mM.

**Table 1 molecules-27-06593-t001:** Results of the purity evaluation of the isolated hepatodamianol using a chromatographic- based procedure and qNMR ERETIC2 tool (based in PULCON method). The % average value of purity is shown, *n* = 3 measurements.

Quantitation Method	% Purity	%RSD
Chromatography	56.98	0.42
qNMR ERETIC2	32.82	2.38

**Table 2 molecules-27-06593-t002:** Longitudinal relaxation time (*T*_1_) results.

^1^H Atom (Substance)	*T*_1_ (s)
CH_3_-6‴ (rutin)	0.43
CH_3_-6‴ (hepatodamianol)	0.53

**Table 3 molecules-27-06593-t003:** Linearity results for ERETIC2 (based on the PULCON method) and external standard methods.

Calibration Method	Curve Equation	Determination Coefficient (r^2^)
ERETIC2	4,208,618.33 x − 455.71	0.99999
External standard calibration with rutin	4,094,508.77 x + 32,601.21	0.99989

**Table 4 molecules-27-06593-t004:** Intraday precision and accuracy results (average results given, *n* = 3) for external standard and ERETIC2 methods.

Theoretical Concentration, mM	ESC ^a^ Concentration, mM	% Error	%RSD	ERETIC2 Concentration, mM	% Error	%RSD ^b^
0.250	0.240	–4.012	0.585	0.241	–3.467	0.633
0.500	0.497	–0.636	0.284	0.491	–1.733	0.311
1.000	1.010	0.961	0.889	0.990	−1.000	0.898
2.000	2.031	1.528	1.214	1.983	–0.833	1.198
3.000	3.012	0.397	0.609	2.938	–2.067	0.621
4.000	3.988	–0.302	0.619	3.888	–2.808	0.616
5.000	4.998	–0.049	0.222	4.870	–2.600	0.229

^a^ ESC, external standard calibration; ^b^ %RSD, percent relative standard deviation.

**Table 5 molecules-27-06593-t005:** Interday precision results (average results given, *n* = 3) obtained by ESC and ERETIC2 (based on the PULCON method).

Concentration, mM	%RSD ^a^ ESC ^b^	%RSD ERETIC2
0.50	1.104	0.311
2.00	1.504	1.198
5.00	0.933	0.229

^a^ %RSD, percent relative standard deviation; ^b^ ESC, external standard calibration.

**Table 6 molecules-27-06593-t006:** Hepatodamianol recovery results (*n* = 3) obtained by ESC and ERETIC2 (based on the PULCON method).

	ESC Quantitation			ERETIC2 Quantitation		
Sample (mg)	Unspiked Sample (mg/mL)	Spiked Sample (mg/mL)	Recovery (%)	Unspiked Sample (mg/mL)	Spiked Sample (mg/mL)	Recovery (%)
Average	1.05	1.42	99.94	1.03	1.39	97.22
%RSD	2.61	0.52	6.36	2.60	0.53	6.32

**Table 7 molecules-27-06593-t007:** The LOQ was established as the concentration with %RSD and % Error < 5 (absolute) in triplicate readings of rutin solutions. The LOQ was determined at 0.250 mM.

Concentration (mM)	%RSD		% Error	
	ESC with Rutin	ERETIC2	ESC with Rutin	ERETIC2
0.125	1.85	2.01	–12.00	–8.27
0.250	0.58	0.63	–4.10	–3.47

**Table 8 molecules-27-06593-t008:** Results of robustness evaluation for ESC and ERETIC2 methods. The evaluated parameters were the following: the LB value, processing (by three different analysts) and baseline correction function. A *p* value < 0.05 was the cutoff to establish significant difference.

			ESC (Rutin)				ERETIC2			
Parameter	Concentration (mM)	Value	Average Concentration (mM)	%RSD	*p*-Value	Robustness	Average Concentration (mM)	%RSD	*p*-Value	Robustness
Processing *	0.5	A1	0.49	0.05	0.281	Yes	0.49	0.00	0.139	Yes
		A2	0.50	0.28			0.49	0.31		
		A3	0.50	0.37			0.49	0.31		
	2.0	A1	2.03	1.36	0.996	Yes	1.99	1.35	0.996	Yes
		A2	2.03	1.21			1.98	1.20		
		A3	2.03	1.12			1.98	1.11		
	5.0	A1	5.14	4.44	0.392	Yes	5.01	4.43	0.391	Yes
		A2	5.00	0.22			4.87	0.23		
		A3	5.00	0.24			4.87	0.24		
LB value	0.5	0.05	0.49	0.21	0.097	Yes	0.49	0.24	0.139	Yes
		0.10	0.50	0.28			0.49	0.31		
		0.15	0.50	0.59			0.49	0.65		
	2.0	0.05	2.03	1.29	0.998	Yes	1.98	1.27	0.998	Yes
		0.10	2.03	1.21			1.98	1.20		
		0.15	2.03	1.20			1.98	1.20		
	5.0	0.05	5.00	0.25	0.709	Yes	4.87	0.24	0.708	Yes
		0.10	5.00	0.22			4.87	0.23		
		0.15	4.99	0.21			4.87	0.21		
Baseline correction algorithm	0.5	ABS	0.50	0.20	0.002	No	0.49	0.20	0.002	No
		ABSN	0.50	0.28			0.49	0.31		
		ABSD	0.49	0.25			0.49	0.24		
	2.0	ABS	2.03	1.31	0.875	Yes	1.98	1.30	0.873	Yes
		ABSN	2.03	1.21			1.98	1.20		
		ABSD	2.02	1.30			1.97	1.31		
	5.0	ABS	5.00	0.26	0.092	Yes	4.87	0.26	0.092	Yes
		ABSN	5.00	0.22			4.87	0.23		
		ABSD	4.97	0.29			4.85	0.28		

* Processing done by three different analysts.

**Table 9 molecules-27-06593-t009:** Concentration of hepatodamianol in standardized extract samples of *T. diffusa* determined by ESC and ERETIC2 (based on the PULCON method). Results are given as the average of three measurements (*n* = 3).

	Hepatodamianol (mg)/Sample (g) (%RSD)	
*T. diffusa* Sample	ESC	ERETIC2
1	36.13 ± 1.17	35.33 ± 3.20
2	52.44 ± 1.37	51.25 ± 2.60
3	28.69 ± 0.39	28.08 ± 1.36
4	36.16 ± 0.84	35.37 ± 2.29

## Data Availability

Not applicable.

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
