# Peer review of "Two Ways to Achieve the Same Goal—Two Validated Quantitative NMR Strategies for a Low-Abundance Natural Product in Standardized Extracts: The Case of Hepatodamianol in Turnera diffusa"

_molecules, 2022, doi:10.3390/molecules27196593_

Round 1

Reviewer 1 Report

The paper is an excellent example of a well - conceived and thoroughly presented differnet methods of quantification of low - abundance secondary metabolites. I have only one question: why did You choose Turnera diffusa as an example plant?

Author Response

Turnera diffusa is a medicinal plant widely used. We found hepatoprototective and hypoglucemic activities. We added an explanatory sentence in the new version (lines 44-48) .One reference (number 14)  was also added.

Reviewer 2 Report

A Brief Summary

The manuscript is well written and has some significant findings. The language is clear and easy to understand. The hypothesis is well stated and clearly defined and, based on current references.

The results are presented in tables and figures very illustrative of the proceedings to compare the method based on a conventional external standard calibration and the second one based on the Pulse Length–based Concentration determination (PULCON) method using the ERETIC2 module as a quantitation tool available in TopSpin software.

The objective of this work is well defined in the final of the introduction, to develop and validate a quantitation method for hepatodamianol in standardized extracts of T. diffusa, using two calibration modalities to compare its performance, PULCON through ERETIC2 77 and the method based in external standard calibration.

The validation of the method was inspected for specificity, linearity, precision (evaluated as %RSD), accuracy (evaluated as the relative % Error and percentage recovery), LOQ, LOD, and robustness, according to the EuroLab guidelines for NMR quantitation.

The paper is well structured throughout, and the conclusions are supported by the results. Overall the study is good and adds something new to the existing literature which may have a positive impact.

The authors concluded that the ERETIC2 represents an effective alternative as a calibration method because saves time and resources, when compared with the calibration by an external standard because is unnecessary, the preparing solutions for the calibration curve.

Author Response

Thank you for your time

Turnera diffusa is a medicinal plant widely used. We found hepatoprototective and hypoglucemic activities. We added an explanatory sentence in the new version (lines 44-48) .One reference (number 14)  was also added

We checked some minor English spelling mistakes and were corrected

Reviewer 3 Report

1H qNMR is highly suitable tool for the simultaneous selective recognition and quantitative determination of metabolites in plant extracts. This work represents the adjustment of two qNMR methods, the ESC and PULCON/ERETIC2, so as to be effectively applied to the quantification of hepatodamaniol, a bioactive component of Turnera Diffusa. This is an example of a comprehensive NMR study representing a great deal of useful experimental information, which can be regarded as a valuable contribution to practical NMR. The MS is well written and it is very clear. I recommend publishing this paper in Molecules, with a few corrections to be made:

1) LL. 83-85: Delete “This section may be divided by subheadings. It should provide a concise and precise description of the experimental results, their interpretation, as well as the experimental conclusions that can be drawn.”

2) Check the spelling of the abbreviation “ERETIC2” throughout the text.

Author Response

  • LL 83-85 were deleted. Our mistake. Sorry 
  • We checked the spelling of ERETIC2 and we found some wrong (Table 1). We corrected them
  • We checked some minor English spelling mistakes and were corrected.